# Subsidence associated with oil extraction, measured from time-series analysis of Sentinel-1 data : case study of the Patos-Marinza oil field, Albania

Marianne Métois[1], Mouna Benjelloun[1], Cécile Lasserre[1], Raphaël Grandin[2], Laurie Barrier[2], Edmond Dushi[3], and Rexhep Koçi[3]

[1]Laboratoire de Géologie de Lyon, Université de Lyon, Université Lyon 1, ENS de Lyon, CNRS, UMR 5276 LGL-TPE, F-69622 Villeurbanne, France
[2]Université de Paris, Institut de Physique du Globe de Paris, UMR 7154, F-75238 Paris cedex 05, France
[3]Institute of Geosciences, Energy, Water and Environnement, Polytechnic University of Tirana, Tirana, Albania

**Correspondence:** Marianne Métois (marianne.metois@univ-lyon1.fr)

**Abstract.** The Patos-Marinza oil field in Central Albania (40.71°N,19.61°E), operated since 1939, is one of the largest onshore fields in Europe. More than 7 millions oil barrels are extracted every year from the Messinian sandstone formations of the Durres Basin in the Albanian Peri-Adriatic Depression by the Bankers company, which has been operating the field since 2004.

In this study, we take advantage of the new Sentinel-1 radar images acquired every 6 to 12 days over Albania to measure the surface displacement in the Myzeqeja plain and in the Patos-Marinza oil field in particular. Images from two ascending and descending tracks covering the area are processed through a radar interferometry (InSAR) time-series analysis over the 2014 to 2018 time-span, providing consistent average Line-Of-Sight velocity maps and displacement time-series.

    The regional deformation field exhibits a slow subsidence of the entire basin relative to the highlands (at rates of 2.5 mm/yr),
that we interpret as a combination of natural and man-induced compaction. This broad picture is complicated by a very strong local subsidence signal with rates as high as 15 mm/yr that spatially correlates with the Patos-Marinza oil field and is maximal in the zone holding most of the operating wells, where Enhanced Oil Recovery techniques are used.

    The striking spatial correlation between the maximum subsidence area and the active wells, as seen from optical images, argues in favor of surface deformation induced by oil extraction. This deformation is well reproduced by elastic models mimick-
ing the basin and reservoir compaction using planar negative tensile (closing) dislocations. Such modeling provides a first-order estimation of the volumetric deflation rate in the oil reservoir (~0.2 Mm$^3$/yr) and suggests that concurrent injection activity has been conducted in the central part of the field where small uplift is observed. Our new InSAR-derived evidences of significant surface strain associated with the oil field operations raise the question of the potential impact of these operations on the local seismicity. A slight increase in the nearby released seismic moment rate seems to be observed since 2009, shortly after the oil
field reactivation. However, without further seismological monitoring of the area and longer InSAR time-series, this question will remain open.

## 1 Introduction

Ground deformation due to oil or gas extraction from buried reservoirs has been reported for over a century. The first dramatic example of such a man-induced subsidence was observed in the Goose Creek oil field, Texas, from 1918 to 1925, with more than 3 meters of vertical motion in the centre of the field (Pratt and Johnson, 1926; Coplin and Galloway, 1999). Since then, significant subsidence patterns have been observed over many productive oil or gas fields (e.g. Schoonbeek et al., 1976; van Thienen-Visser et al., 2015; Grebby et al., 2019), so much so that they are now routinely monitored by the operating companies (Nagel, 2001). Indeed, since surface deformation is mostly due to reservoir depletion and compaction of the sedimentary layers, monitoring is a convenient indirect way to constrain the reservoir pressure and volume evolution through time. Soil and surface instability caused by extraction can, however, substantially damage buildings and infrastructures (e.g. Doornhof et al., 2006). Furthermore, drilling processes and fluid extraction can change the pore pressure and stress field in the medium sufficiently so that non-tectonic seismic activity may be triggered (stress changes of 0.1 to 1 MPa are sufficient, e.g. Segall, 1989; Segall et al., 1994; Ottemöller et al., 2005; Rubinstein and Mahani, 2015; Hettema et al., 2017; Foulger et al., 2018). Tracking surface deformation in the context of oil extraction is therefore crucial to assess direct and indirect hazards connected to industrial activity.

In the last decades, the search for ever increasing well productivity led to novel recovery methods based on horizontal drilling, hydraulic fracturation, or deep injection of fluid or steam to drive the oil out of the reservoir. The efficiency of these Enhanced Oil Recovery (EOR) techniques comes with larger and faster changes in the stress field at depth, increasing their potential for triggering local anomalous seismicity (Murray, 2013; Rubinstein and Mahani, 2015). The fact that extensive extraction and injection of fluids in the subsoil can trigger moderate-sized earthquakes is thus now commonly accepted, in particular where gas is extracted (Foulger et al., 2018, and references therein). The HiQuakes database of potentially induced-earthquakes built by Foulger et al. (2018) hosts 38 earthquakes associated with EOR techniques in general and with waterflooding in particular, as observed for instance in the Wilmington and Inglewood oil fields, USA, in the 20th century (Nicholson and Wesson, 1992). Unravelling the contributions of (i) injection for EOR, (ii) oil extraction itself, or (iii) wastewater disposal in inducing seismicity and deformation is challenging since these operations are often occurring simultaneously. One of the largest induced events ever recorded is the 2016, $M_w$ 5.7 Pawnee earthquake, that ruptured a previously unmapped basement fault under the Oklahoma oil and gas field and was probably induced by wastewater disposal (Murray, 2013; Walsh and Zoback, 2015). This event suggests that anthropogenic subsurface activities can influence the stress state over a broad region both at depth and laterally (Keranen et al., 2013; Grandin et al., 2017), increasing substantially the regional seismic hazard (van Elk et al., 2017). However, monitoring the stress and pressure perturbations associated with an operated hydrocarbon field and understanding their relationship with the background seismicity require very dense seismic arrays, continuous monitoring of deformation and a good knowledge of the reservoir characteristics (geometry, pressure history, etc). Unfortunately, this information is seldom available.

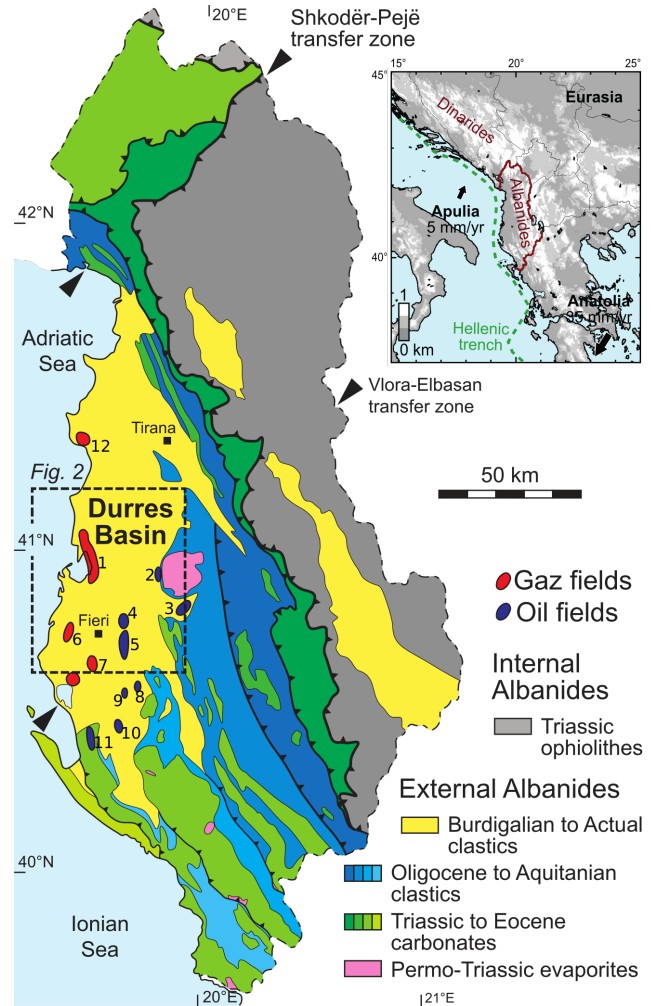

**Figure 1.** Simplified geological map of Albania. Oil and gas main fields are indicated in blue and red, respectively and numbered : 1-Ballaj-Divjaka, 2-Pekishti, 3-Kucova, 4-Bubullima, 5- Patos-Marinza, 6-Seman-Poveica, 7-Frakula, 8-Balshi, 9-Cakrani, 10-Gorishti-Koculi, 11-Drashovice, 12-Durresi. Dashed rectangle is the study area as plotted in Fig. 2. Upper right inset shows the geographical location of Albania and regional kinematics with respect to stable Eurasia (Métois et al., 2015).

Albania has been a long-standing hydrocarbon producing country since natural bitumen surges were already exploited during Roman times (Verani and Ineichen, 1942). Effective oil drilling started at the beginning of the 20th century, taking advantage of the large deposits of gas and crude oil accumulated in the Ionian carbonates and siliclastic deposits of the Peri-Adriatic Depression in the Durres Basin in Albania (Figures 1 and 2 and Sejdini et al., 1994). In the Patos-Marinza oil field (40.71°N,19.61°E) operated since 1939, heavy crude oil contained in shallow (few hundreds to 2000 m depth) Messinian siliclastic reservoirs has been extracted since 2004 by the Canadian firm Banker's petroleum (Fig. 3). To increase the productivity of this very large onshore oil field previously exploited using primary recovery techniques during the last century, EOR techniques including

waterflooding, infill, thermal recovery and horizontal drilling have been applied to the northern part of the field, close to the Marinza village, starting in 2008 and leading to spectacular productivity gains (Figures 3, 4 and BCP, 2015). Since then, several events have raised concern among the local population about the risks associated with such an intensive extraction activity (Fig.4-b). In 2013, three $M_w$~4 earthquakes that occurred in the area alarmed the population, that was already claiming that the overall background seismicity had increased since at least 2012 (CAO complaint). On 1 April 2015, the Marinza village was evacuated due to leakages of natural gas during drilling operations that contaminated water wells (Contamination report, 2015). More recently, in December 2016, a shallow seismic swarm (maximum $M_L$ 4 at 2.9 km depth) developed, leading the government to open a public inquiry and enjoining Bankers to stop the water injection in the area (Report CAO, 2018).

The lack of a dense seismic network operating in the area over this period and the scarce knowledge of the pressure history of the reservoir prevent scientists from concluding a potential causality between the oil extraction activity and the current seismicity (Report CAO, 2018). However, the monitoring of surface deformation in the oil and gas exploitation area may at least give clues on how the strain and stress field is affected by this exploitation. In this study, we take advantage of the dense spatio-temporal coverage of the new Sentinel-1 satellite radar images that span the Patos-Marinza region, with one image every 6 to 12 days. We use these images to build consistent maps of the surface displacement over the 2014 to 2018 time span, using the Interferometric Synthetic-Aperture Radar (InSAR) technique. We then describe the subsidence and uplift patterns observed over the Patos-Marinza oil-field, quantify the associated motion rates, and suggest that such patterns are likely related to reservoir draining and associated compaction.

## 2   History, geology and seismotectonic setting of the Patos-Marinza oil field

In Albania, the oil and gas fields are concentrated in the external Albanides, composed of Mesozoic limestones that have been thrusted, eroded and partially covered by Oligocene to Pliocene marine and deltaic sedimentary series accumulating in the Peri-Adriatic Depression (Fig. 1 and Meço et al., 2000). The emerged part of this depression, i.e. the Durres Basin, hosts most of the productive fields in Albania, among which the Patos-Marinza field this study focuses on (Fig. 2). The shallowest formations encountered in this southern part of the Durres Basin are Holocene and Pleistocene lagoonal, marshy and alluvial deposits (Fig. 2-c) related to important variations in depositional environments of the Myzeqeja plain and its shoreline (Ciavola, 1999). Today, the plain is drained by both the Shkumbini and Semani rivers, often flooding and changing their channel paths, and by a dense network of man-made canals built during the 1950–1980 period of reclaiming works for agriculture and public health purposes (Fig. 2-b, Shallari and Maughan, 2015). To the west, the basin is hardly above sea level and has deviated from its sedimentary equilibrium due to the recent anthropogenic reclamation. The coastline has thus drastically changed in the last decades (Ciavola and Simeoni, 1995; Shallari and Maughan, 2015), leading to a need for building artificial dikes. Apart from agriculture, oil and gas extraction is the main economic income of the Fieri prefecture.

In Albania, the Patos-Marinza oil field is singular. First, this is one of the oldest oil fields in operation as it started being operated in 1939 (Fig.3-a) and has supported the Albanese industry during the 20th century. Second, the heavy crude oil extracted there comes from shallow molassic layers while most of the other Albanian oil fields operate in deeper Ionian

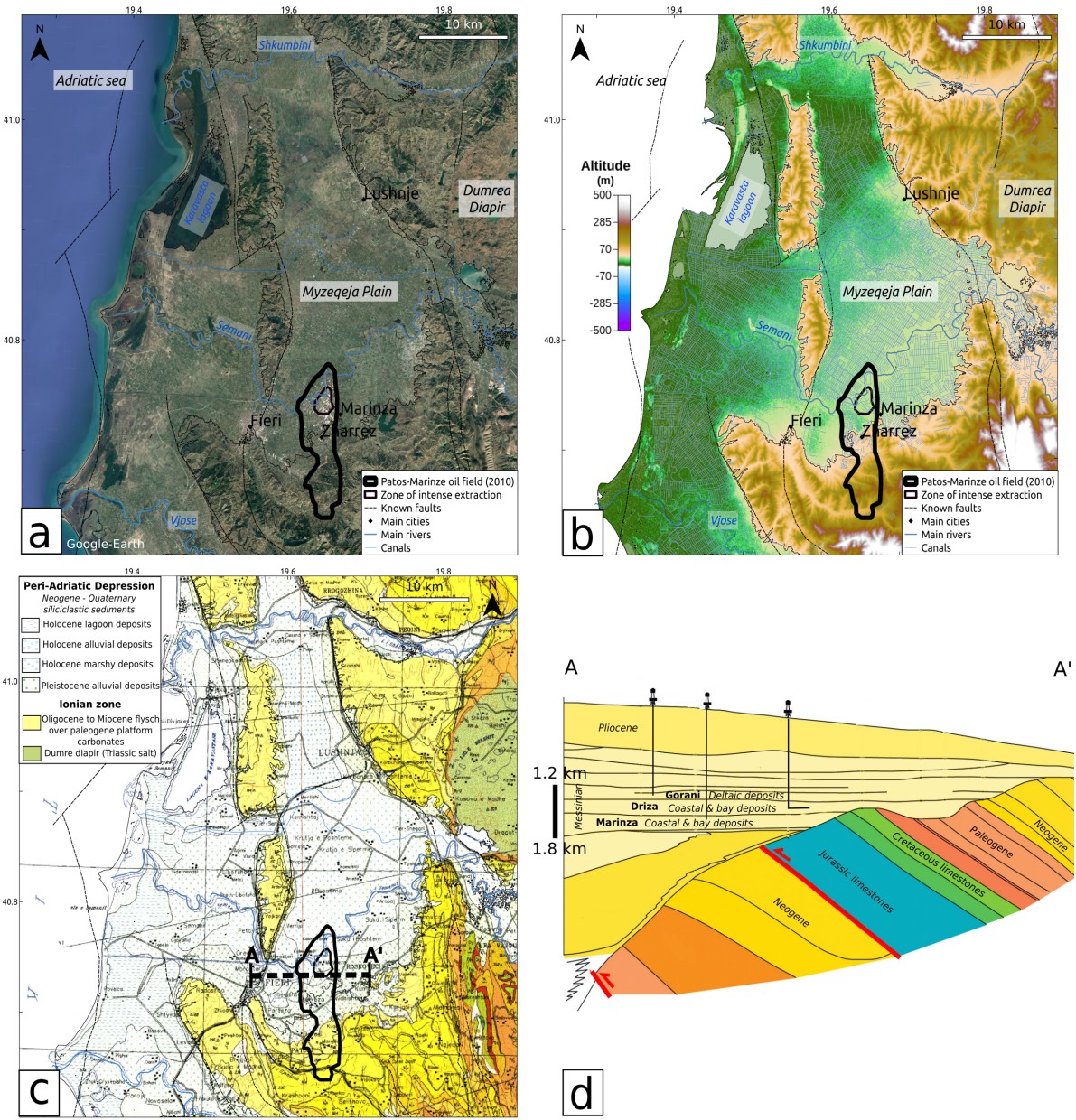

**Figure 2.** Anthropogenic, hydrological and geological context of the study area. The Patos-Marinza active oil-field (black bold line) is contoured based on BCP, 2015. It encompasses the zone where wells are the densest and extraction activity concentrates (thin black line). Faults (black dashed lines) are from Aliaj et al. (2000). a-Optical satellite image of the southern part of the Durres Basin in the Fieri prefecture (satellite images from the Google Earth data base). b-Hydrology of the Myzeqeja plain. The network of man-made canals (in blue) is from TPGINC and is superimposed to the topography of the area (Farr and Kobrick, 2000). c-Extract of the Geological Map of Albania (simplified from Xhomo et al., 2002). The dotted bold line is the approximate position of the simplified geological transect presented in the panel d. d-West-East interpreted geological cross-section modified from Silo et al. (2013). The average depth of the Messinian sandstones oil-bearing formations drilled in the northern part of the Patos-Marinza oil field is indicated together with the associated depositional environments. Faults in the underlying sedimentary series are plotted in red.

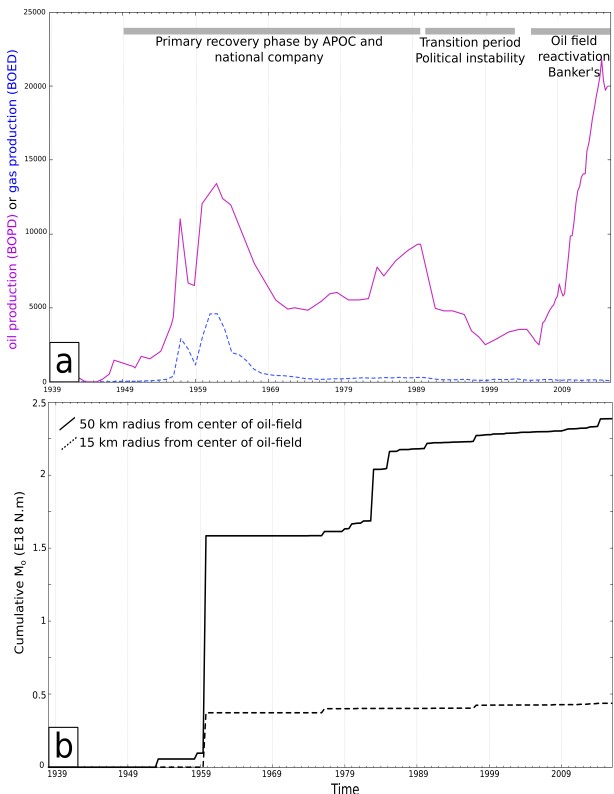

**Figure 3.** a- History of production of oil (plain line in Barrel Oil per Day) and gas (dashed line, in Barrel Oil Equivalent per Day) in the Patos-Marinza field (BCP, 2015; IHS, 2019). b- Cumulative seismic moment released since 1939 based on a combined USGS-EMSC catalog (USGS; CSEM-EMSC, before and after 2004, respectively) over a 50 km radius area (plain line) and a 15 km radius area around the oil field (dashed line). The contours of these zones are plotted in figure 5.

limestone reservoirs (Meço et al., 2000). The oil extracted in the Patos-Marinza field has migrated from the Ionian Mesozoic source rocks into ~10°NW dipping Messinian sandstone reservoirs, located at depth ranging from few hundreds of meters to roughly 2 km (Fig. 2-d, Silo et al., 2013; Prifti and Muska, 2013). While the first production wells were located south of the oil field (south of 40.7°N) in the so-called Patos zone, most of the currently operated wells are located north of 40.7°N, in

5 the flatest part of the field called the Marinza zone. The core of the Patos-Marinza oil field covers a ~45 km$^2$ zone over the Marinza and Zharrez municipalities in the Fieri prefecture. It is characterized by a very high spatial density of wells visible on satellite optical images (white areas in Fig. 2-a). There, three units are currently exploited, mostly between 1200 and 1800 m depth (Weatherill et al., 2005), from top to bottom : the Gorani, Driza and Marinza series (Fig. 2-d), with various porosities (between 25 and 30%), compaction coefficients, and chemical and physical characteristics of the oil in place (Bennion et al.,

10 2003). These series are therefore exploited using different extraction techniques (Fig. 3-a). The Albanian national oil company first operated the field for both oil and gas using primary recovery methods until 1999, when secondary recovery techniques, in particular Cold Heavy Oil Production by Sand (CHOPS), started to be used mainly in the Driza unit (Fig.3-a and Sejdini

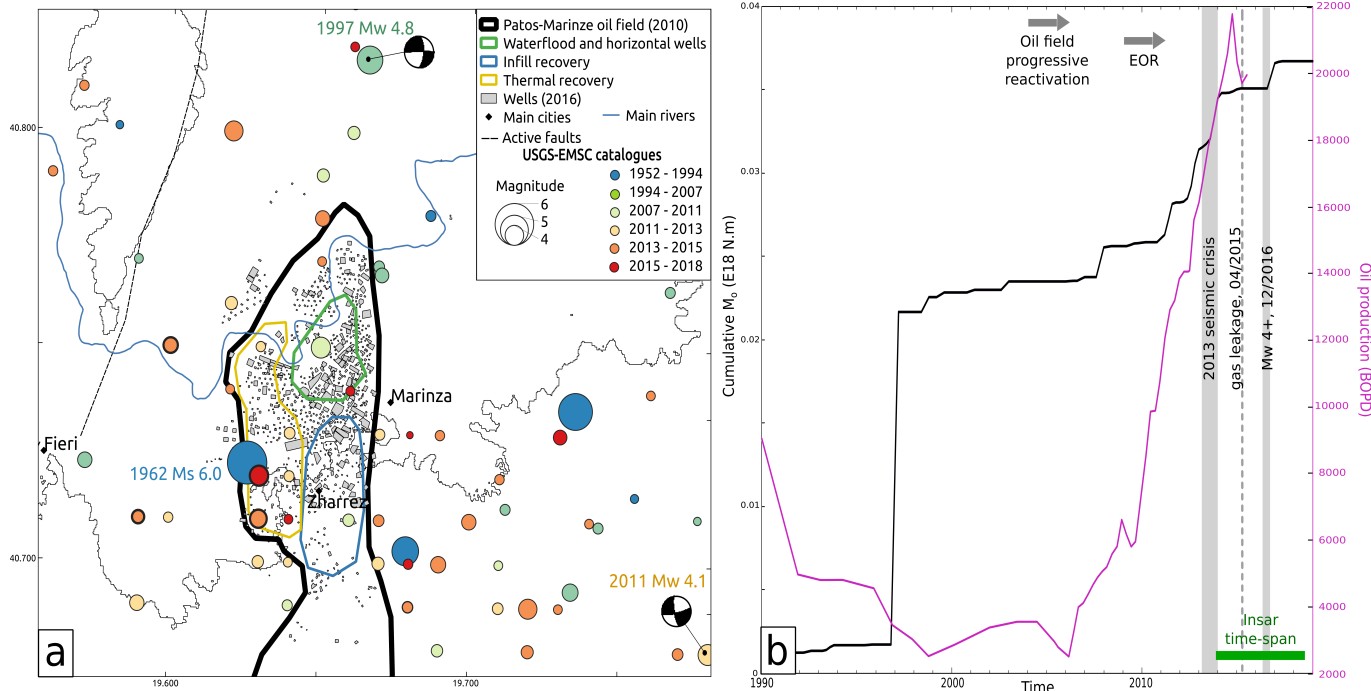

**Figure 4.** a- Exploitation extent and recent history of the Patos-Marinza oil field based on BCP, 2010. Major wells (grey rectangles of various sizes) have been mapped based on 2016 optical image (Fig. 2-a and satellite images from the Google Earth data base). Seismic events since 1950 are color-coded based on their occurrence time and size depending on their magnitude (USGS; CSEM-EMSC, before and after 2004, respectively). Earthquakes of the 2013, Mw 3.5+ sequence are shown by circles contoured in bold black, as well as the 2016, Mw 4+ event, all cited in the text. b- Production evolution in Barrel Oil per Day (1 BOPD ~159 L) since 1990 (BCP, 2015) together with the cumulative released seismic moment based on USGS and EMSC catalogues for a 15 km radius small region around the center of the oil field.

et al., 1994; Weatherill et al., 2005). Because of the high viscosity of the oil in place (30 to 3000 mPa.s Weatherill et al., 2005), EOR methods have then been applied to the field by the Bankers company since 2009 in order to increase the production. While infill recovery is applied preferentially to the deepest Marinza unit, the Driza unit is now exploited using waterflooding, horizontal drilling and thermal recovery (Fig. 4 and BCP, 2015). As a result, more than 20,000 Barrel Oil Per Day (BOPD)
5    were extracted from the oil field in 2014 (Fig. 4-b). The waterflooding area (green contour in Fig. 4-a) has also been the first zone to be reactivated in 2005 (BCP, 2015) and is the part of the field with the highest number of active wells, as mapped in Figure 4 using optical images from 2016. It is therefore reasonable to consider this part of the field as the most intensely operated today, and as so it will be contoured for reference in the following figures (thin black line).

Because it is located between the Hellenic subduction zone and the Dinarides collision belts (Fig. 1), Albania is an earthquake-
10    prone country that often experiences moderate earthquakes and has already suffered from devastating $M_w$6+ earthquakes (see Jouanne et al., 2012; Métois et al., 2015, and the 2019, Mw 6.4 Durres event). The external Albanides deform due to the collision between the ~5 mm/yr NE-moving Apulia microplate and the Balkan Peninsula migrating south toward the Aegean

Sea (Fig1). They are characterized by thrust and strike-slip focal mechanisms attributed to active NNW-SSE thrust and fold structures and NE-SW trending transform zones (Aliaj et al., 2010; Jouanne et al., 2012). Large historical earthquakes are suspected to have occurred in the past in the Fieri area based on narrations of the antique Apollonia destruction (II-III BC, 234 A.D Shebalin et al., 1998). A large $M_S$ 6 earthquake also stroke the city in 1962 (Aliaj et al., 2010; Jouanne et al., 2012, and Fig. 4-a). Two focal mechanisms are provided in the area by the CMT catalogue for $M_w$4+ earthquakes that occurred at 6 and 10 km depth for the southern (2011) and northern (1997) events, respectively (Fig. 4-a). Both earthquakes exhibit clear strike-slip motion that is consistent with the reactivation of faults affecting the Ionian limestone units. However, due to inadequate spatial coverage by seismic arrays, the precise depth of these events remains uncertain.

Between 2004 and 2017, 93 shallow earthquakes (above 15 km depth) occurred less than 15 km away from the Marinza village, among which 5 were $M_w$ 4+. In particular, a $M_w$ 4.1 event occurred in December 2016 west of Zharrez, very near to the 1962 epicenter (Fig. 4-a). This event, which increased the residents' concern, led the government to order a temporary pause in the injection activities that were suspected of inducing seismicity (Report CAO, 2018). Over this period, the EMSC magnitude of completeness for the whole country is around 3.5 (Fig.S1), i.e. still too high to conduct a proper analysis on whether the observed seismicity could be induced (in the broad sense of "related to human activities in general", Foulger et al., 2018) or not.

However, as a first element of discussion, and keeping in mind that the seismic record in our study area may be incomplete for the pre-2000 period due to the sparse seismic network at the time (see Fig.S1 for the temporal evolution of the magnitude of completeness), we calculate the cumulative seismic moment evolution starting in 1939 and 1990, based on the combination of the USGS (before 2004) and the EMSC (after 2004) catalogues, for large (50 km radius, encompassing the whole Myzeqeja plain) and small (15 km radius) areas around the oil field (Fig. 3-b). Because we are interested mainly on the small to intermediate magnitude events, we consider earthquakes with magnitude ranging from 3.5 (completeness magnitude) to 6. Overall, no clear increase in the rate of released seismic moment can be observed for both regions since the 1962 Ms 6 event (Fig. 3-b), rulling out the hypothesis of a primary influence of the oil production on the regional seismicity. However, for the post-1990 period in the 15 km around the oil field, a slight increase in the cumulative seismic moment may be identified (Fig.4), in particular starting in 2008. Although this recent increase remains too small to be significant based on the longer term trend, its coevalness with the increase in the Patos-Marinza oil production likely explains the raising concern of the neighboring population.

Again, investigating on the causal link that may exist between the oil extraction activity and the local seismicity would require a fine monitoring of the background seismicity in this tectonically active region over a long period. Unfortunately, this information is currently lacking. Following a complaint filed by the population to an independent office (the Compliance Advisor Ombudsman, CAO) after the 2013 seismic crisis (Fig. 4 and Report CAO, 2018), two broadband stations have been installed by researchers from University Polytechnics of Tirana in the central part of the oil field starting in September 2016, in order to record the intense seismic activity of the zone. However, because only two seismometers were installed, earthquake locations for small magnitude events are subject to large uncertainty and the observations have remained inconclusive up to now (Report CAO, 2018).

Geodetic techniques represent another indirect method to monitor the effect of oil field exploitation. We present below an InSAR analysis of surface ground motion in the Patos-Marinza field area to investigate the potential link between the field exploitation and the present-day surface deformation.

## 3   InSAR data and processing

We use data from the European Space Agency (ESA) Sentinel-1 Synthetic Aperture Radar (SAR) constellation, consisting of two satellites launched respectively in April 2014 (S1A) and April 2016 (S1B). Sentinel-1 operates in a burst mode (the so-called TOPS mode, De Zan and Guarnieri, 2006), which allows for generating images ~ 250 km wide across-track, acquired in three contiguous sub-swaths. Each Sentinel-1 satellite operates on a sun-synchronous orbit with a cycle time of 12 days. The two satellites are phased at 180°, enabling a revisit time of 6 days for interferometry. According to the acquisition plan designed

by ESA, systematic acquisitions are carried out over Europe, including Albania, providing an optimal temporal coverage for our area of interest. We process independently SAR data acquired on two overlapping tracks in descending (T153) and ascending (T175) geometries, consisting respectively of 120 and 109 images over the 2014–2018 period (Figs. S2, S3).

Sentinel-1 data processing is carried out using the NSBAS software (Doin et al., 2011), which includes routines from the ROI_PAC software (Rosen et al., 2004). The originality of NSBAS is to enable the application of a series of corrections prior

to phase unwrapping, in order to minimize unwrapping errors, which remain the primary limitation of the InSAR techniques in areas of low coherence. The specific Sentinel-1 processing technique used in NSBAS is described in Grandin (2015) and Grandin et al. (2016). After selecting the common bursts in our image set, a master image is chosen on the basis of its central position in terms of spatial baseline and time of acquisition within the image stack (Figs. S2, S3). We use the 30 m SRTM digital elevation model (Farr and Kobrick, 2000) to compute the simulation in the master geometry. Then, all slave images are

coregistered to the master geometry using an affine transformation constrained by incoherent pixel correlation with the master image, and their simulated phase is computed using precise orbital information.

A network of small baseline interferograms (Berardino et al., 2002) is selected, minimizing spatial baselines, while retaining variety in temporal baselines and sufficient redundancy. A total of 512 interferograms are calculated on the descending track (Fig S2), and 468 interferograms on the ascending track (Fig S3). Precise azimuthal coregistration is achieved by computing

burst-overlap interferograms using the spectral diversity technique (Grandin et al., 2016). The residual azimuth offsets with respect to the master image are first evaluated using spectral diversity independently for each interferogram. They are then inverted within the interferogram network, in order to ensure consistency and mitigate potential ambiguity for poorly coherent interferograms. The resulting interferograms on adjacent sub-swaths are subsequently merged by solving for an integer number of $2\pi$ shifts between the sub-swaths. We checked for the absence of phase jump across the sub-swaths borders and between the

bursts.

Finally, atmospheric correction using the ECMWF's ERA-Interim atmospheric reanalysis is carried out using the method described in Jolivet et al. (2011) and Doin et al. (2009). After this correction, remaining atmospheric delays correlated with to-pography are further estimated using an empirical linear relation between phase and altitude, and consistently inverted within

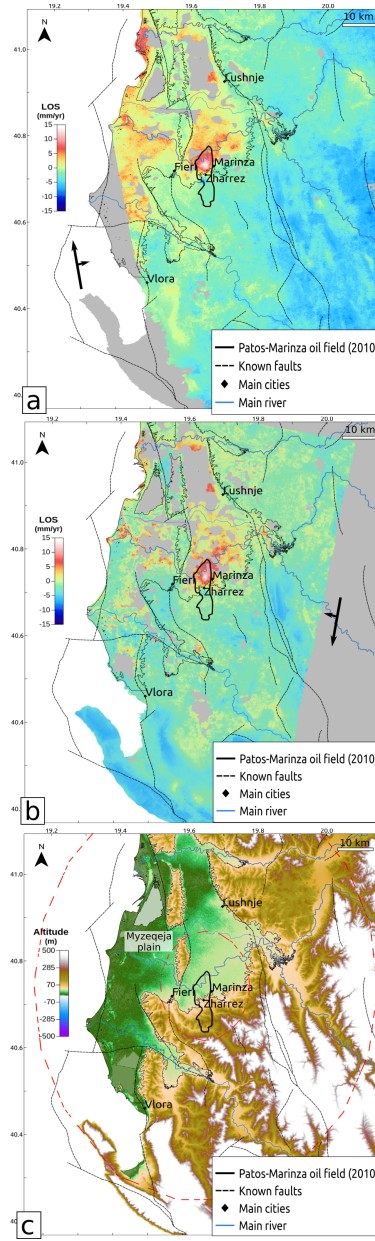

**Figure 5.** Large-scale view of the Line-Of-Sight (LOS) average velocity maps (positive values correspond to motion away from satellite, i.e. mostly subsidence), from time-series analysis of InSAR data along ascending (a) and descending (b) tracks, together with regional topography (c). The Patos-Marinza contour is plotted fromBCP, 2010.Red dotted ellipses are the 15 km and 50 km radius area around the center of the oil field used for moment rate calculation in figures 3 and 4.

the network. Similarly, large-scale phase ramps in range, likely caused by residual large-scale atmospheric artifacts, are removed in a network-consistent manner. The resulting interferograms are filtered using a multiscale boxcar filter implemented in NSBAS. Phase unwrapping is also performed using a specific routine of NSBAS by propagating the unwrapping integration path across the interferogram, starting from a coherent reference pixel, and expanding progressively into areas of lower coherence (Grandin et al., 2012).

After unwrapping, all interferograms are registered to a reference area and a time-series inversion is carried out, in order to solve for the temporal evolution of ground displacement between each acquisition date, and for the average velocity of each pixel independently (Figs. 5 to 7). Residual unwrapping errors are solved as part of the time-series inversion using an iterative technique (Cavalié et al., 2007, and references therein).

The median value of the interferometric network misclosure (root mean square in Fig. S4) is 0.5 rad and 1 rad for the descending and ascending tracks, respectively. We choose to mask pixels on velocity maps (Fig. 5) based on the number of interferograms used to estimate the average velocity. The minimum number of coherent interferograms required for each pixel is fixed to 370 for each track.

## 4  Results

We extract the average linear Line-Of-Sight (LOS) velocity from the raw displacement time-series of both ascending and descending tracks (Fig. 5) by simultaneously estimating for each pixel the amplitude of annual and semi-annual sinusoidal signals using the least-squares inversion strategy of Daout et al. (2017). The ascending track exhibits higher levels of annual variations than the descending one, in particular at the coast and at high altitudes (Fig. S5). This may result from a significantly higher tropospheric variability at the acquisition time of the ascending track (6:33 PM local time, i.e. dusk), whereas acquisitions in the descending track take place in more quiet atmospheric conditions (6:45 AM local time, i.e. dawn). However, in general, the level of annual and semi-annual variations is very low for both tracks over the Patos-Marinza oil field (Fig. S5).

The average LOS velocity maps shown in Figure 5 for both tracks exhibit very consistent deformation patterns and are highly correlated (r=0.97, Fig. S7). This similarity implies that the surface displacement is weakly dependent on the LOS vector and is therefore mainly vertical. In the following, we will interpret the LOS velocities as nearly vertical motion, neglecting the small horizontal component. For further comparison purposes, we remove a uniform constant (3.24 mm/yr) from the descending LOS velocity map (as determined in Figure S7) to match it with the ascending one. We then calculate the covariogram for each velocity map, based on a stable zone covered by both tracks south of the Myzeqeja plain (Fig. S7). Both maps have a similar variance for zero distance but covariance is slightly higher at intermediate distances (10 to 20 km) for the descending map than for the ascending one. This is consistent with the patchy signal that can be observed at this kilometric scale in the descending velocity map only (Fig. 6).

The primary feature of these surface velocity maps is the subsidence of the whole Myzeqeja plain (Fig. 2, relative motion away from the satellite of ~2.5 mm/yr in average) with respect to the surrounding highlands, which show no significant deformation (Figs. 5 and 6). However, subsidence is not uniform over the alluvial and deltaic basin and higher subsidence rates can

be locally observed (Fig. 6) (i) along the Semani River (5 to 10 mm/yr in LOS direction), (ii) west of Lushnje (~10 mm/yr), (iii) at the mouth of the Shkumbini River (~10 mm/yr) and (iv) concentrated in a 2.1 km by 2.1 km area in the northern part of the Patos-Marinza oil field with the highest rates (up to 20 mm/yr). We also identify a small uplifting area (~5 mm/yr in the LOS direction), roughly circular, immediately south of the maximum subsiding area and of the Zharrez village (Figs. 6 and 7).

Figure 7 shows the displacement time-series of these areas of maximum subsidence and uplift in the oil field with respect to a stable zone : both trends are very linear on the 2014-2018 time-span, with no jump associated with the main events that affected the oil field during this period (gas leakage and $M_w$4+ earthquake), or oscillations that could be related to the seasonal or annual hydrological loading. The higher temporal resolution (return time of 6 days) available in the area since the end of 2016 will help track small and sudden deviations from the average trend in future studies. We limit our analysis below to that
of the average velocity field.

## 5   Discussion and modeling

### 5.1   Subsidence of the Myzeqeja plain

The striking dichotomy between the slowly subsiding Myzeqeja alluvial to deltaic plain (~2.5 mm/yr, Fig. 6) and the stable surrounding calcareous and molassic highlands argues in favour of a compaction phenomenon, widely observed in poorly
consolidated alluvial and deltaic systems similar to the Shkumbini and Semani ones (e.g. Liu et al., 2004; Higgins, 2016, and references therein). Several natural processes may be involved, among which compaction of clay layers due to the sediment net flux, isostatic adjustment, or seasonal and annual elastic or poro-elastic rebound due to external loading. Human activities may also amplify the natural subsidence of alluvial and deltaic zones in several ways: (i) pumping of groundwater or other buried fluids can accelerate the compaction process (e.g. Liu et al., 2004), (ii) changes in the hydrographic network can dramatically
modify the streamflow paths, the water discharge and the net flux of sediments, and (iii) dense urbanization, as an external load, can favor compaction (e.g. Abidin et al., 2011).

   All of these subsidence-accelerating human actions are at play in the Myzeqeja plain context. First, underground water coming from alluvial shallow aquifers is pumped out for irrigation purposes and drinking water networks. Second, the marshy areas were recently reclaimed during the 1945 to 1980 period by building an extensive network of canals (Fig. 2-b, Shallari
and Maughan, 2015). At last, the construction of dams upstream the Shkumbini and Semani rivers, of hill retention structures, and of breakwaters at the coast, have modified the sedimentary flux balance to the point that dramatic changes in the coastline have been observed between both rivers mouths (Ciavola and Simeoni, 1995; Ciavola, 1999; Bedini, 2007). Unravelling the different natural or anthropogenic sources of the plain subsidence is an arduous task and would require a fine knowledge of the water discharge, sediment budget, and mechanical behavior of the shallow sediment layers (Higgins, 2016). This problem is
beyond the scope of this study. However, we suggest that the observed basin-wide subsidence may result from a combination of natural and human-induced compaction. In particular, we interpret the high subsidence values observed along the Semani River and the Shkumbini River mouth as the consequence of the repeated deviations of their riverbeds and changes in the incoming sedimentary flux (Ciavola, 1999). A large part of the basin has been masked west of Lushnje given our confidence criteria after

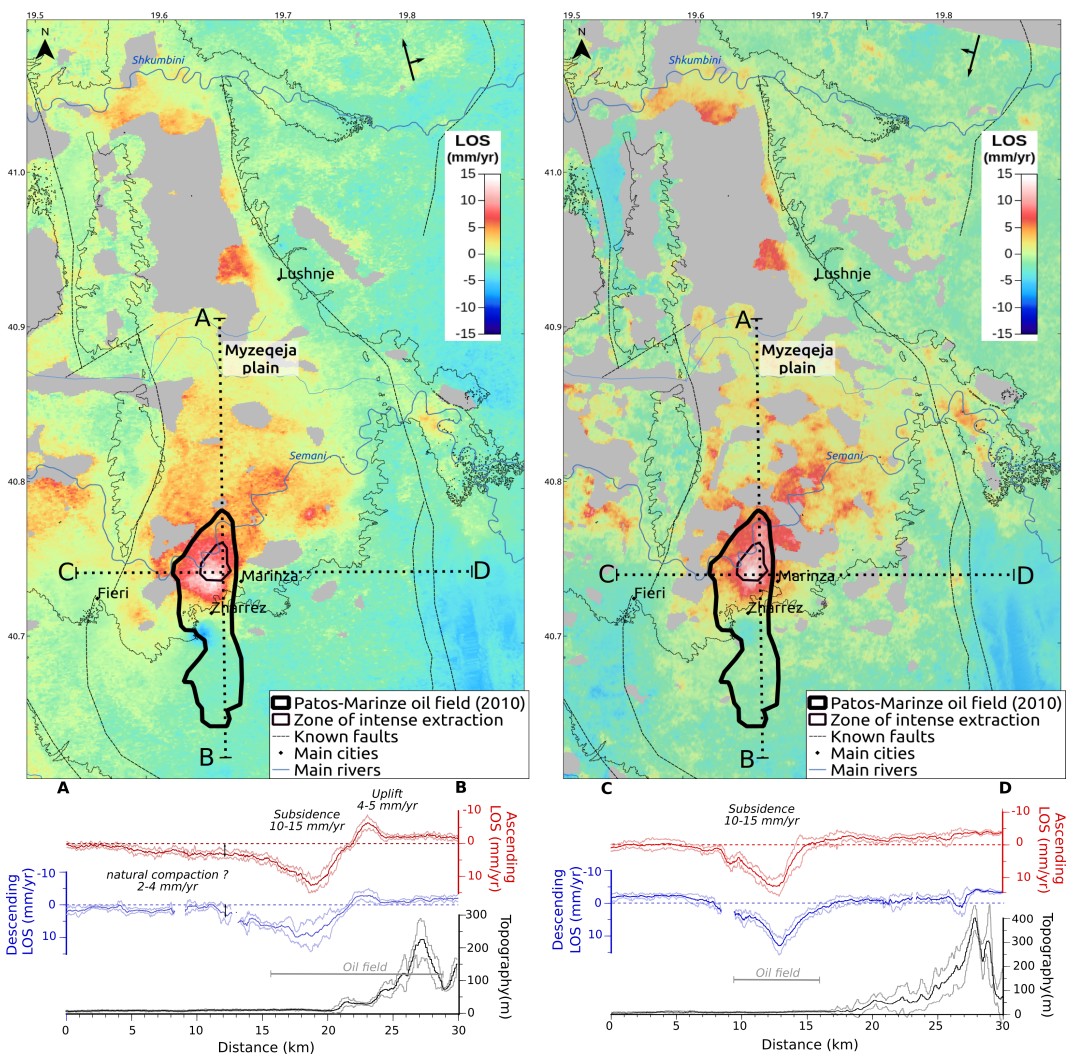

**Figure 6.** Top: Zoom on the LOS average velocity map for the ascending (left) and descending (right) tracks over the Patos-Marinza oil field. Color code and sign convention are the same as in Figure 5. Dotted lines indicate the location of the profiles at bottom. Bottom: north-south (left) and west-east (right) velocity profiles across the oil field's most intense deformation zone together with topography profiles (bottom). Average values over a 2 km width across the profile lines is plotted darker, while lighter lines limit the $2\sigma$ standard deviation envelop.

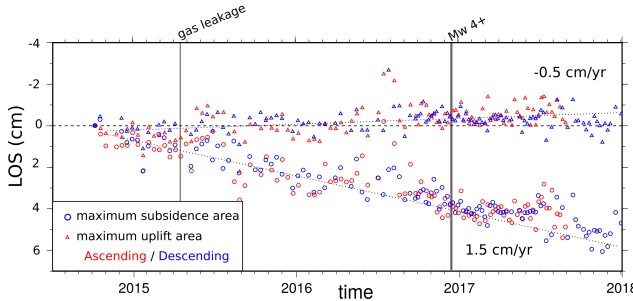

**Figure 7.** Displacement time-series for the most subsiding and uplifting areas (3x3 pixels) on both tracks (ascending is red, descending is blue) in the Patos-Marinza oil field, relative to a stable area. The average trend is indicated (dashed line), together with two particular events (grey vertical bars) related to the field exploitation and seismic activity.

time-series analysis (see section 3), but some rapid subsidence is still observed around this area. Since this zone is slightly under sea level and has long remained marshy (Ciavola, 1999; Shallari and Maughan, 2015, and Fig. 2), we interpret this local high subsidence rate as resulting from natural compaction associated with recent land reclamation. However, since the 4-year time-span of the InSAR study remains relatively short compared to the characteristic time-scale for long-term compaction of

sedimentary basins (e.g. Chaussard et al., 2013), a longer time-series would be required to further characterize the specific behavior of the Myzeqeja plain.

## 5.2 Deformation in the Patos-Marinza field

The LOS deformation pattern observed over the Patos-Marinza oil field differs in many ways from the diffuse subsidence of the Myzeqeja plain described above (Fig. 6). Looking at this mainly vertical deformation along north-south and west-east

LOS velocity profiles highlights again the excellent consistency between both tracks. The spatial correlation between the fast subsidence zone and the active part of the oil field, intensely operated using waterflooding recovery also emerges clearly (Fig. 4). The slight uplift observed further to the south is located at the southern boundary of zones where thermal and infill recovery methods have been used according to BCP, 2010. This uplift is roughly circular and is very similar to deformation patterns that would be associated to local inflation of a buried source (Mogi, 1958; Lisowski, 2007). The shape of the area of fast subsidence

appears also nearly radial on the west-east profile but is superimposed with the overall subsidence of the alluvial plain on the north-south profile (Fig. 6). The nearly axi-symmetric shape of the observed surface deformation pattern, together with the clear spatial correlation between high subsidence rates and the zone where intense oil extraction is conducted, argue in favor of a local subsidence induced by reservoir compaction as observed elsewhere (e.g. Chaussard et al., 2013; Liu et al., 2015; Grebby et al., 2019).

As stated in the introduction, many other smaller oil and gas fields are currently operated in the Durres basin (Fig.1). We therefore check whether a similar subsidence pattern could be observed over the fields covered by our velocity maps (Fig.S8). No significant surface deformation can be identified, suggesting that the Patos-Marinza context is unique. A possible hypothesis

for this strong deformation occurring in this intensely operated oil field could be that gas may still be retrieved coeval with extraction of heavy oil, leading to pressure drops larger than expected in the reservoir. An important gas leakage has indeed been observed near the most subsiding area in 2015 (Report CAO, 2018).

Surface deformation has been monitored over a wide range of oil and gas fields worldwide, and several physical models have been proposed and validated with real data (e.g. van Thienen-Visser and Fokker, 2017). In the simplest elastic case, analytical solutions developed for buried points or finite spherical sources (Mogi, 1958), tensile dislocations (Okada, 1985), and circular sill-like horizontal cracks (Fialko et al., 2001), can provide first-order models for surface deformation due to buried volumetric or pressure changes. Analytical formulations based on the poroelastic theory have also been proposed in the case of a compacting cylindrical reservoir to calculate the associated stress and displacement field in the medium (Geertsma et al., 1973; Segall et al., 1994; Goebel et al., 2017). The surface deformation field predicted by all of these physical models can differ significantly in the horizontal and vertical components (Lisowski, 2007). However, discriminating between these different physical models requires both horizontal and vertical components of surface deformation or the reservoir geometry to be known, which is not the case in our case study. Indeed, we lack information on the reservoir characteristics, and InSAR is mainly sensitive to vertical motion (Dieterich and Decker, 1975; Lisowski, 2007; Fialko et al., 2001). We therefore adopt a two-step approach to model at the first order the velocity field observed in the eastern Myzeqeja plain (Fig. 6).

First, we use the poroelastic formalism developed for an axisymmetric compacting reservoir by Segall (1989) to conduct a 1D simple inversion of the west-east LOS velocity profiles for both tracks presented in Figure 6. We invert for the depth, pressure change, and radius of the compacting reservoir using simple least-square optimisation. We fix the compaction coefficient to $3.10^9 \mathrm{Pa}^{-1}$, the Poisson ratio to 0.25, and we impose the reservoir to be 600 m thick based on Weatherill et al. (2005). This thickness is fixed because of its trade-off with the pressure drop. We also impose the center of the reservoir to be located under the maximum subsidence area. Observed LOS velocities are well reproduced by a 1045 m-long reservoir ($\pm$80 m) located at a depth of 1.6 km ($\pm$0.1 km) under the zone of maximum subsidence in which pressure change is around 32.7 kPa/yr ($\pm$7 kPa/yr, see fit to the data in Fig. 8). These parameters are consistent with the average depth and extent of the Driza and Marinza series that are the most intensely exploited Miocene units in the Patos-Marinza oil field (Fig. 2-d). The observed surface subsidence would therefore correspond to a decrease of reservoir volume $\Delta V$ of roughly 0.2 Mm$^3$/yr on the 2014-2018 time-span. The subsiding part of the north-south LOS velocity profiles presented in Fig. 8-C is fairly well explained by a poroelastic deflating reservoir presenting these characteristics. However, such a model logically fails in retrieving the smallest subsidence rates observed north of the field and the slight uplift south of it.

We therefore propose to use the parameters adjusted using the previous simple poroelastic inversion to invert for the 2D velocity field using a simpler elastic formalism. We thus discretize the alluvial plain into 2.1 km×2.1 km horizontal tensile dislocations located at 1.6 km depth based on the poroelastic modeling (Fig. 8). Because of the scarce information available about the reservoir geometry, we chose to use horizontal dislocations as a first approximation. It is also to note that because the dislocation depth is of the same order of magnitude as its side width, we are at the limit of validity of the elastic approximation (Lisowski, 2007). The dislocations located outside the oil field will mimic the natural compaction effect described in section 5.1 that homogeneously affects the plain: the amount of tensile slip (opening or closing) on these dislocations (in meter) is

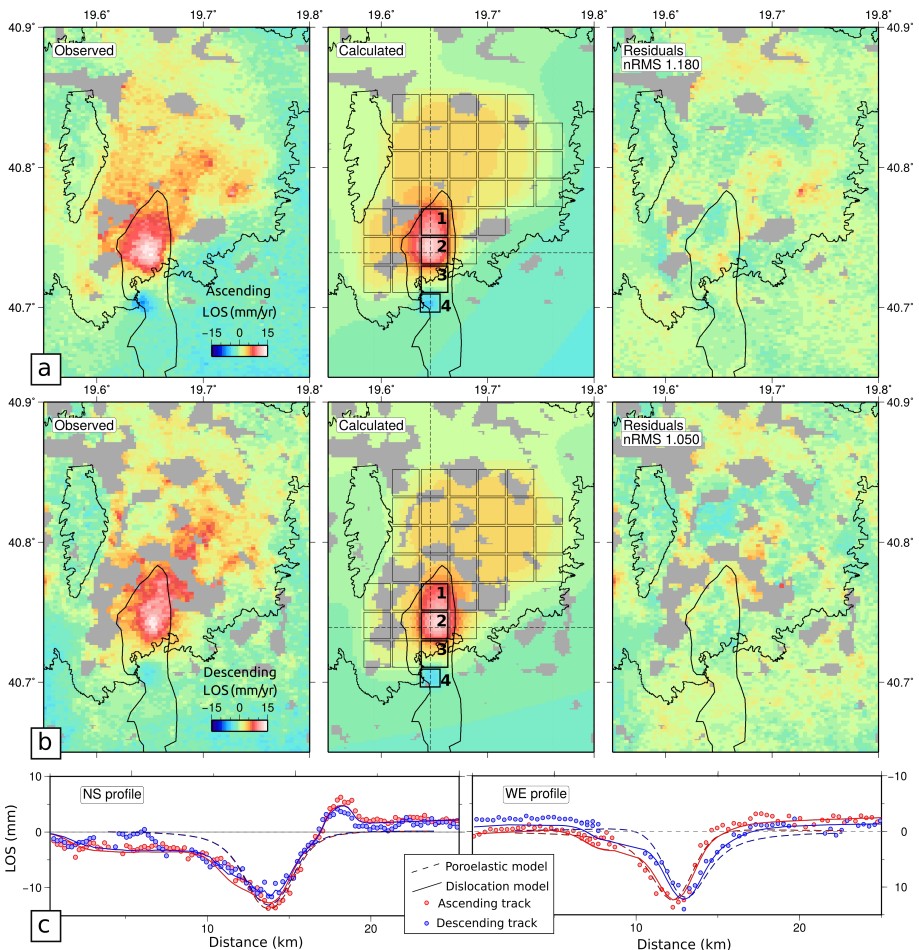

**Figure 8.** a,b- Calculated LOS velocity maps (left) are compared with predicted ones, produced by planar crack Okada-type dislocations in a homogeneous elastic half space (centre), for both the ascending (a) and descending (b) tracks. Residuals are plotted on the right panel. Grey dashed lines in central panel show location of profiles at bottom. Numbered dislocations contoured in bold are those inside the oil field (see text for details). c- north-south and west-east profiles with observed (dots) and predicted (lines) velocities using either the planar-cracks model presented above (plain line) or a 2D poroelastic axysymmetric reservoir (dashed line) (see text for details). Velocities on the ascending and descending tracks are shown in red and blue. Note that the 2D poroelastic model predicts nearly LOS-independent velocities in the NS direction.

| Dislocation | Depth (km) | Surface (km$^2$) | Tensile slip (m) | $\Delta V$ (Mm$^3$/yr) |
|---|---|---|---|---|
| 1 | 1.6 | 4.4 | **-18±0.5** | **-0.079** |
| 2 | 1.6 | 4.4 | **-27.3±0.6** | **-0.120** |
| 3 | 1.6 | 4.4 | **-4±0.4** | **-0.016** |
| 4 | **0.5±0.2** | **2.2±0.3** | **+5.4±1.7** | **+0.015** |
| 5-36 | 1.6 | 4.4 | **-5.6±0.07** | **-0.025** |

**Table 1.** Parameters used in the best-fit multiple tensile dislocations model presented in Fig. 8. Dislocations inside the most intensively exploited part of the oil field are numbered from 1 to 4. Parameters adjusted during the inversion are plotted in bold characters together with their uncertainties. We convert the amount of tensile slip on each dislocation (column 4) into an equivalent volume (column 5) to compare with the results from the poroelastic modeling.

therefore forced to be homogeneous over the entire basin (see Table 1). On the contrary, the amount of opening/closing of the dislocations located inside the most intensively exploited part of the oil field (numbered from 1 to 4) are independent parameters. Since the spatial wavelength of the uplifting signal is significantly smaller than the breadth of the subsiding area, we allow the depth of dislocation number 4 over the uplifting area to vary between 0 and 4.5 km. We use the Tdefnode code

developed by McCaffrey (2009) to invert for the amplitude of tensile slip on each dislocation as well as the depth and size of dislocation 4. The code is also free to search for a constant and linear ramps in the EW and NS direction to adjust tightly the two inverted velocity maps taking into account the local LOS vectors. In total, 14 free parameters are adjusted.

This multiple planar dislocations model produces a non radial deformation pattern that fits very well both observed velocity maps with normalized Root Mean Square (nRMS) close to 1 (Fig. 8). We summarize in Table 1 the parameters used to build the

final model presented in Figure 8. The dislocations located outside the oil field all experience low amplitude compaction that results in an overall 2.5 mm/yr motion in the LOS. Accordingly, this approximate modeling approach is efficient to mimic the overall basin subsidence but is not physically consistent with the actual compaction mechanism that is likely more distributed over the sedimentary pile. We therefore refrain from interpreting the volumetric changes associated with these dislocations and rather focus on the oil field deformation modeling.

Dislocations 1 to 3 experience compaction equivalent in total to a reservoir deflation of ~0.215 Mm$^3$/yr, i.e. of the same order of magnitude as the volumetric change found using a 1D poroelastic model. The only dislocation experiencing inflation (at a rate of 0.015 Mm$^3$/yr) is dislocation number 4, located south of the Zharrez village where uplift is observed. It should be noted that this source is found to be very superficial and that the volume involved is more than one order of magnitude lower that the total volumetric deflation found in the intensely exploited region. We interpret the circular uplift located south of the

Zharrez village as an indication of local fluid injection activity in the shallow part of the sedimentary pile that may be due to leakage of injection well or to wastewater injection associated with oil extraction. Indeed, localized uplift associated with these activities has already been extensively described (e.g. Teatini et al., 2011).

Keeping in mind the limitations of the elastic assumptions to model surface deformation due to shallow reservoir deflation and the fact that our model parametrization do not take into account the complex geometry of the reservoirs, our findings suggest that the observed surface deformation pattern over the Patos-Marinza oil field is consistent with compaction of the Gorani, Driza and Marinza reservoirs. The estimated deflation rate is relatively low (~0.2 Mm$^3$/yr) compared to the average volume of oil extracted on an annual basis since 2014 (~1.16 Mm$^3$/yr, Fig. 4). This apparent discrepancy can be explained by several factors. First, the oil production is calculated by the operating company over the entire oil field. Therefore, this calculation takes into account oil that has been extracted everywhere in the field with different extraction techniques or in other reservoirs. Second, in the most intensely operated zone experiencing the highest subsidence rates, oil is extracted via waterflooding techniques (Fig. 4) that imply substantial injection of fluids at depth. The net volumetric change experienced by the reservoirs is therefore lower than what would result from oil depletion alone (Pierce, 1970). To properly assess the net volumetric change of the reservoirs, one would need to build a 3D finite element model of the media taking into account its physical properties, geometry, pressure history, etc., i.e. data we do not have access to at the moment.

## 6 Conclusions & implication for local seismic hazard

In this study, we use an InSAR time-series analysis to document a previously undescribed subsidence pattern over the most intensely exploited zone of the Patos-Marinza oil field with large subsidence rates (of up to ~15 mm/yr), which stands out from the otherwise slowly subsiding Myzeqeja plain (at a rate of ~2.5 mm/yr). We interpret this locally high subsidence as due to man-induced compaction associated with the use of EOR techniques in the shallow Driza, Gorani and Marinza heavy oil reservoirs. We also document a slow uplift rate in the center of the oil field (up to 5 mm/yr) that could be associated with localized wastewater disposal.

Such type of surface deformation can be associated with stress changes in the neighboring geological formations, which have been correlated with low to intermediate magnitude seismicity in several well-instrumented oil and gas fields (e.g. Segall et al., 1994; Ellsworth, 2013; Keranen et al., 2013, 2014; Hornbach et al., 2016). Induced seismicity is now routinely monitored by operating companies, together with the pressure evolution of the reservoirs and the pressure history of each injection well, in order to avoid sharp stress changes that may favor earthquake triggering. Because they involve large amounts of injected fluids at depth for long periods, wastewater injection and EOR techniques are particularly prone to inducing seismicity (Rubinstein and Mahani, 2015; Foulger et al., 2018). The fact that we do observe significant surface deformation colocated with the active part of the Patos-Marinza oil field over the 2014-2018 period raises the issue of whether the reservoir compaction may also induce seismicity during this period. Unfortunately, local seismological arrays are still too sparse to provide accurate locations and focal mechanisms of the small magnitude earthquakes occurring in the vicinity of the Patos-Marinza oil field, a necessary pre-requisite to conclude on the induced nature of the recent apparent increase in local seismicity (Fig. 4). Since Albania is an earthquake-prone country where seismic hazard is high (Jouanne et al., 2012; Métois et al., 2015), discriminating between tectonic-related and man-related seismic events there remains challenging and out of the scope of this study.

The occurrence of the $M_w$ 5.7, Pawnee earthquake, which broke a previously unknown basement fault under an intensely exploited hydrocarbon field in Oklahoma (Keranen et al., 2013; Grandin et al., 2017), illustrates the fact that stress changes associated with subsurface human activities, in this case wastewater disposal, may affect a broad area and trigger significant earthquakes. It remains, however, unclear whether human activities are more susceptible of triggering earthquakes in tecton-

ically active environments or in intraplate contexts (Göbel, 2015). Nonetheless, we would like to point out the fact that the Ionian limestones that form the basement of the Myzeqeja plain have been largely folded and thrusted and are still tectonically active due to the ongoing compression in the external Albanides (Jouanne et al., 2012; Guzmán et al., 2014; Aliaj et al., 2000), as underlined by the 2019, Mw 6.4 Durres earthquake. In this context, the possible interference of human activity with an otherwise significant seismic hazard deserves a particular attention. Whether the stress changes associated with the oil extrac-

tion in the Patos-Marinza field may be sufficient to trigger earthquakes on underlying faults is therefore an open question that challenges the seismic-hazard assessment in the area. Denser and more precise seismic catalogues, together with longer InSAR or other geodetic time-series monitoring the spatio-temporal evolution of the deformation, as well as detailed knowledge of the wells injection history and reservoir properties, are essential to explore this issue further on.

*Author contributions.* M.B., C.L and R.G processed the radar images and produced the InSAR time-series. M.B. and M.M. modelled the

data. L.B., R.K and E.D gave insights on the geological, seismological and industrial context. M.M coordinated the ALBA project and the redaction of the present paper. All authors participated in the interpretation and discussion of the results.

*Competing interests.* The authors declare no competing interests.

*Acknowledgements.* This research is part of the ALBA project funded by the CNES APR program. Significant support in the French-Albanese collaboration has been provided by the Université de Lyon Impulsion-Palse program granted to M.Métois. InSAR data were

processed using the NSBAS chain, mainly developed at ISTerre and IPGP as part of a french community effort to provide tools for the Data & Services center for Solid Earth (ForM@Ter). The figures of this manuscript were done using the free and open source GMT library (http://gmt.soest.hawaii.edu/) and the QGIS software (http://qgis.osgeo.org). The authors would like to thank one anonymous reviewer and the editor S.McKlusky for their comments that helped improving the manuscript, and T.Shreve for proofreading this paper.

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
