# Peer review of "Subsidence associated with oil extraction, measured from time-series analysis of Sentinel-1 data : case study of the Patos-Marinza oil field, Albania"

_Solid Earth, 2019_

## Referee Comment (RC1) · Anonymous Referee #1 · 12 Sep 2019

Review of " Subsidence associated with oil extraction, measured from time-series analysis of Sentinel-1 data : case study of the Patos-Marinza oil field, Albania" by Metois et al.

The paper provides original data and analyses on ground deformation contemporary to a shallow oil field exploitation.ÂăÂă The paper focusses on the analysis of (2014-2018) surface deformation in the vicinity of the Patis-Marinza oil field, Albania, that operates since 1939. The main result points on an average 15 mm/yr subsidence rate, 2014-2018, above the reservoir zone "where most of the horizontal wells are located".

[Figure]

Alternatively, crude use of seismological catalogs unbalances the detailed technical analysis for the interferograms. The well-constrained technical results for subsidence are discussed against poor seismicity and production history and without the proper fluid manipulation context, concerning the cited literature (most cited references relate to seismicity driven by wastewater disposal which is not the case in the submitted manuscript).Ăă ĂăThe lack of production information and the misused of seismicity catalogs weaken the central message of the manuscript. Most of the discussions on the possible anthropogenic seismicity are flawed in several ways. First, there is no information on how the regional earthquake catalog is built on or selected by merging existing catalogs (USGS and EMSC). Second, the lack of threshold values (i) for selecting an earthquake at a given distance from the reservoir to be a triggered event or a tectonic event and (ii) for a magnitude completeness value over time, prevent any robust analysis of space-time seismicity patterns.Ăă From another perspective, the comparison of seismicity and deformation with long term production history is missing. The current 15mm/yr subsidence should drive a more than one-meter subsidence in the past 80 years if sustainable over time. Another possible change point may be used as before-after 2009 when Enhanced Oil Recovery processes started. The one-meter subsidence in the past 80 years should be evidenced locally independent from any detailed instrumental subsidence survey. It will put in context if the current subsidence rate was stable over time or if there is an increase in the recent deformation relatively to past subsidence. These deformation patterns (steady-state or not) are critical parameters for forecasting the future deformation of the area.Ăă Furthermore, one advantage of the interferometry analysis is the ability, by the same tools, to sample a large area. Accordingly one would expect the authors to take advantage of the several oil and gas fields that are located in this area (see fig. 4) to calibrate and validate the surface deformation and seismicity related to these different hydrocarbon production sites.Ăă Accordingly, I recommend the paper to be deeply reconstructed. For one option, the authors may focus on the interrelationships between subsidence and regional hydrocarbon extraction on different sites. To investigate the possible relationship between

seismicity and hydrocarbon production in this zone, it requests new and robust inputs for the seismic analyses. Âă Other technical comments and suggestions are listed below.

Specific comments : Âă

oil field operations: - In several places in the manuscript (including the abstract) the authors point on the subsidence bowl to be centered above the area where horizontal production wells are located. There is neither discussion nor reference to support why horizontal production wells should increase the subsidence. We can find in existing reports (not cited in the manuscript) a one order magnitude change in production rate on that period. Such a more quantitative information should be discussed in the context of subsidence rate change over time. Âă - In the introduction, it is difficult to understand why the 2008 reactivation of the field may enhance the local seismicity so far away from the 1939 onset time of production. I suggest the authors summarise comprehensively the history of production that is missing in the manuscript. There is information available in open access reports that should be listed in the manuscript, e.g., "the 6% of reserve being produced during the 1939-1990 period, Weatherill et al. 2005 SPEÂăhttps://doi.org/10.2118/97992-MS"; and "The Patos Marinza oilfield peak production during this period was 15,000 bbl/day in 1975 but due to a lack of maintenance, reduction in drilling activity, and lack of new technology the oilfield declined to producing almost nothing in the 1990's and it appeared to be on its last legs. Since 2008, a new Canadian owner has drilled more than 600 horizontal wells and has also implemented enhanced oil programs which have pushed the production to 21,000 bbl/day which is significantly higher than any time in the oilfields history" (Delmaide, 2017)";ÂăÂă - Also, it appears there are problems with how the oil reservoirs are described in the paper (100m-2 km depth) when the available literature points on "Measured drilling depths are typically 2,500 meters, with 500- to 700-meter horizontal legs." (e.g., Mazerov, 2011). Also, Weatherill et al. (2005) report on "The two separate fields (Patos in the south and Marinza in the north) have productive sands at different depths, i.e., 0 - 1200 m for Patos and 1200 - 1800 m for MA". These differences are not discussed in the manuscript.Âă - Hydrocarbon field locations and numbers for other nearby oil-gas reservoirs do not overlap with other existing maps ( e.g., Mezini and Musai, 2012, Velaj SPE, 2015). Information for location and site references are welcome.Âă - The contour for the zone of intense extraction should appear on fig3a to be able to describe how it overlaps with the contoured types of production stimulations

seismicity: Abstract: "the increase of background seismicity": Usually, for seismology community the background seismicity is related to the steady-state tectonic event rate; here we do not know how field operations can increase the background seismicity because there is no analyze to either define or to quantify the seismicity before or far-field from the oil extraction onset or oil reservoir, respectively. l3,vp5: "In 2013, three Mw∼4 earthquakes that occurred in the area alarmed the population". The distances to the oil field reservoir are not described for the 3 events. Furthermore, it is difficult to read who are these events in the figure (3a) where most of the 2013 events are far outside the reservoir contour. Again the description of the rules to accept or to reject the seismic event to be anthropogenic or tectonics is not explicitly described in the manuscript.ÂăÂă - seismicity catalogs: the only information on the seismicity database is provided in figure 3 caption. "CSEM-EMSC; USGS, before and after 2004, respectively". However, using the USGS catalog after 2004 (as downloaded Sept2019), the date and magnitude of the M4+ event in late 2016, that is attributed to the oil field extraction, does not exist on the USGS database even as M2.5+ event? - the significant M4.8 event of figure 3a does not emerge on figure3b using seismic moment value, whereas it is one of the most significant events in this selected time-space window?Âă

subsidence: - Figure 4: what does mean "oil field (2010) contour "on this figure is not defined.Âă - the oil field contour on fig 3 does not match the other figures? - Strong subsidence is located close to the north-western Balla Divjaka gas field, and it is not discussed in the text. -l25-30,p13: "Both ascending and descending velocities are well reproduced by a 1045 m-long and 632 m-wide reservoirs located at a depth of ∼1.6

km under the zone of maximum subsidence". We will appreciate error bars on the meter accuracies of the equivalent reservoir dimensions. Also, if these values are not a-priori imposed, we need to know which ranges were tested for length, width, thickness during parameter tuning of the model. If not, we do not understand what the key outputs of the model are?  earthquake triggering and Patos-Marinza fluid manipulations: - l10,p2: "Furthermore, drilling processes and fluid extraction can change the pore pressure and stress field in the mediumÂăso muchÂăthat anomalous seismic activity may be triggered". This point is out of context in the case of seismicity triggered by oil and gas recovery. One of the main lessons of anthropogenic seismicity related to hydrocarbon recovery is that earthquakes are triggered by small stress change, i.e., a few 0.1-01 MPa, the same order of magnitude of stress change than the ones that trigger aftershocks (e.g., for a review Foulger et al. 2018)Âă -l15,p2: "For instance, the seismicity observed in the Oklahoma oil and gas field and its relationship with wastewater disposal has been extensively studied in the last years (Murray, 2013; Walsh and Zoback, 2015". Here there is a problem because the authors use examples of wastewater injections, that is not similar to enhanced recovery technique, to support the triggering potency of enhanced recovery.Âă - Because no threshold for magnitude completeness is used, the analysis of seismicity patterns over space and in relation to subsidence and production are not robust. Any evidence for an increase in seismicity rate has to be tested not to be biased by a change in the earthquake recording thresholds (e.g.,ÂăJ Woessner,ÂăS WiemerÂă- Bulletin of the Seismological . . ., 2005). The lack of Mmin choice induces biases in seismicity rate changes as driven by change is instrumental network geometry. For all standard seismological studies, the regional choice for Mmin is derived from the frequency-size distribution of earthquakes. ÂăFor the anthropogenic event selection, a proper way to define anthropogenic seismicity at a given reservoir distance is to map different distance ranges and to extract up to which distance the seismicity differs from the background tectonic rate. As an example using the USGS database 1973-2018, M>=4 or M2.5, there is no evidence for earthquake clustering in the vicinity of the oil field (figure R1-2)?Âă - Most of the cited literature

relates to the case for wastewater injection (or gas withdrawal). There is no evidence for the wastewater injection to drive seismicity in this oil field. In the case of the shallow oil field, the initial pressure allows for smaller pressure drop than for gas fields, and it may be the reason why there is four times less reported cases for earthquake triggered by oil rather than gas extraction, respectively (e.g., for a review Foulger et al. 2018). A section that compares subsidence around the Patos-Marinza oil fields with other cases where oil extraction triggered subsidence and seismicity is missing.Âǎ As example l20, p16 "Such type of surface deformation is likely associated with stress changes in the neighbouring geological formations, which have been correlated with low to interme-diate magnitude seismicity in several well-instrumented oil fields (e.g. Segall et al., 1994; Ellsworth, 2013; Keranen et al., 2013, 2014; Hornbach et al., 2016)." These four references are cited to support surface deformation and in-depth stress changes induced by oil recovery., but they are all out of context. Segall et al. 1994, reports on a 4-5 km depth, 65 MPa initial pressure, the gas field associated with a few cm sub-sidence cases, M4 seismicity. Ellsworth 2013 review paper and Keranen et al., 2013 and Hornbach et al., 2016 papers for Oklahoma and texas respectively, all the three reports entitled "Injection-induced earthquakes", do not describe any subsidence data or extraction value, and focus on wastewater injection below reservoirs for which no data are available on the case study of Patos-Marinza fluid manipulation.Âǎ

Figure R1: Seismicity map around the Patos-Marinza oil fields. M>=4 earthquake, 1973-2019, from USGS catalogue. Black box is the schematic contour of the Patos and Mazrinza oil-fields. (annex)

Figure R2: Seismicity map around the Patos-Marinza oil fields. M>=2.5 earthquake, 1973-2019, from USGS catalogue. Black box is the schematic contour of the Patos and Mazrinza oil-fields. (annex)

[Figure]

**Fig. 1.** Figure R1: Seismicity map around the Patos-Marinza oil fields. M>=4 earthquake, 1973-2019, from USGS catalogue. Black box is the schematic contour of the Patos and Mazrinza oil-fields.

[Figure]

**Fig. 2.** Figure R2: Seismicity map around the Patos-Marinza oil fields. M>=2.5 earthquake, 1973-2019, from USGS catalogue. Black box is the schematic contour of the Patos and Mazrinza oil-fields.

---

## Author Comment (AC1) · 23 Oct 2019

We would first like to thank reviewer 1 for his comments and suggestions related to our work on the subsidence of the Patos-Marinze oil field.

As noted by the reviewer, the main result of this paper is a map of LOS (i.e. nearly vertical) surface velocity based on an InSAR time series analysis of Sentinel-1 radar images acquired on the 2014-2018 time-span, that covers the entire Myzeqeja alluvial plain. This is the first InSAR study in this area. Our analysis focusses on a particularly

strong subsidence signal (>1.5 cm/yr) that happens to be concentrated on the northern part of the Patos-Marinza oil field where the oil extraction is currently the most intensely conducted. Our intention in this paper is to bring new and original observations of undocumented subsidence that may constitute a serious societal issue. Therefore, most of the paper is focused on the method used to obtained the LOS time-series, its description, analysis and modeling. We note that no criticism is formulated on the InSAR analysis or modeling by itself in the review, and we conclude that the characteristics of the subsiding signal that we document are not disputed.

On the other hand, the reviewer points that (i) surface deformation should be analyzed on other regional hydrocarbon fields, (ii) the oil field history is too briefly described and the deformation history too short, and (ii) Âń crude use of seismological catalogs unbalances the detailed technical analysis for the interferograms Âż. In the following, we address these concerns and propose substantial modifications to the current manuscript : we propose to add supplementary figures, modify fig.2 and 3, and develop section 2 in order to give a broader view of the context of the study. We will clarify the discussion section and make clear that our conclusions point toward an induced subsidence that is most probably recent and does not extend in time, at least at the present-day rate. We will also clarify the fact that because of the sparseness of the seismic catalogs and of the too recent seismological networks available for the area, we do not claim to conclude on whether induced seismicity did occur in response to oil extraction, but we note that this assertion cannot be disproved either. Again the originality of our study is to map and quantify a previously unknown rapid subsidence over the largest oil field of Albania, that remains poorly studied in the literature.

[Figure]

**1 Surface deformation over other regional hydrocarbon fields**

As stated in the main text, external Albanides host many active oil and gas fields that are roughly represented in Fig.1. Before focusing on the high subsidence pattern observed in the interferograms in the Patos Marinza area, we indeed had a look at the surface deformation in all other oil and gas fields.

As evidenced in figures 1 and 2 of this answer, no comparable motion could be observed in the other oil or gas fields in the area. In particular, the strong subsidence ( 1 cm/yr) observed west of the Balla-Divjaka gas field and noted by the reviewer is not related to oil extraction near Divjaka but we interpret it as associated with the Skumbini delta subsidence (we will make this clearer in the new version of the manuscript). This interpretation is consistent with the fact that this large subsidence pattern is distant by more than 5 km from the gas field itself, and with the description of major reorganization of the deltaic system there (Ciavola and Simeoni, 1995; Ciavola, 1999; Bedini, 2007). We reckon that this useful information is currently lacking in the manuscript and could be added either by incorporating one of the attached figures to Fig.4 or to add both figures as supplementary information.

We agree with the reviewer's statement that subsidence and induced seismicity are usually more often observed above gas fields than above oil fields, because the pressure jumps at depth are larger and because of increased compressibility of the fluids. However, since we do not observe any clear sign of subsidence in the albanian gas fields, nor in the other active oil fields, the Patos-Marinze context may be unique. We recognize that, at this stage, we have no explanation for this apparent peculiarity of the Patos oil field (an hypothesis is that both oil and gas are extracted there, see section 2 below).

We propose to add in the context and discussion parts a brief description of the other oil and gas fields and to make clearer that only the Patos-Marinza filed is affected by significant ground deformation.

**2 Oil extraction and deformation history**

*"From another perspective, the comparison of seismicity and deformation with long term production history is missing. The current 15mm/yr subsidence should drive a more than one-meter subsidence in the past 80 years if sustainable over time. Another possible change point may be used as before-after 2009 when Enhanced Oil Recovery processes started. The one-meter subsidence in the past 80 years should be evidenced locally independent from any detailed instrumental subsidence survey. It will put in context if the current subsidence rate was stable over time or if there is an increase in the recent deformation relatively to past subsidence. These deformation patterns (steady-state or not) are critical parameters for forecasting the future deformation of the area."*

We agree with the reviewer that, since the oil field has been operated for decades, it would be very interesting to look at surface deformation on a time-frame longer that what we currently propose in the paper, which represents only a recent snapshot of ground deformation (i.e. LOS velocities on the 2014-2018 time span), highlighting significant localized subsidence.

However, if a leveling network exists in Albania since 1860 and has been densified, remeasured and upgraded since then, we do not have access to the data (see http://asig.gov.al/images/Relacioni%20permbledhes.pdf or Nikolli Idrizi, 2011). One easiest way to go back further in time and better constrain the history, and thus the controlling parameters of the observed subsidence, would be to process older InSAR data sets such as ALOS images that are available for the 2006-2011 period, or the older ERS-Envisat archive from 1991 to 2011. This will indeed be done in the coming years but requires a huge processing work and is beyond the scope of this study. Nevertheless, we believe that our study may be commended for documenting for the first time this recent and significant subsidence spot.

This being said, we can imagine some speculative scenarios on the possible evolution

of the subsidence history in the area, based on the scarce but existing knowledge of the oil production history. The current version of the paper (from line 21 p.4 to line 4 p.6) proposes a brief overview of the oil production history based on the available references [Weatherhill, 2005, and Bankers Corporative Reports, 2010, 2015]. We agree that this description could be refined and more detailed, and we will do so in the next version of the manuscript.

In particular, we should indeed underline the fact that the Patos-Marinza field has been indeed producing oil and gas simultaneously for many years, before oil production became largely dominant. As pointed out by the reviewer, gas extraction usually goes together with larger pressure drops in the reservoirs in comparison with oil extraction. We could therefore expect that, if surface subsidence is indeed caused by reservoir compaction as we do suggest, it may have been significant from 1958 to ∼1963 during the maximum gas production period (see figure 3 of this answer). This point will be added to the manuscript and we will discuss an hypothesis that we have previously discarded : that the localized subsidence seen in the northern part of the Patos-Marinze oil field could be associated with local gas extraction that is still ongoing (we recall that there was a gas leakage in 2015 in the zone).

Second, the Patos-Marinze field is composed of numerous reservoirs. In the main text, we focus on the soft siliclastic shallow reservoirs of the Driza, Gorani and Marinza suites that appear to be currently the most intensely operated, though limestone reservoirs have also been, and are still, operated. Obviously, compaction would be higher in the siliclastic reservoirs than in limestones.

Finally, as shown in figure 3 of this answer that corresponds to an enlarged view of figure 3b of the main text, the production history of the oil field is complex and unsteady. In particular, after the primary recovery period, the production dropped in the 90's and nearly stopped in 1999. If our hypothesis of extraction-induced compaction is valid, we doubt that the subsidence rate observed on the 2014-2018 time span could have remained stable for 80 years. We agree that this discussion is of interest for the readers

and propose to : - provide in the text a more exhaustive history of the oil field production based on the available literature (it is to note that the bibliography on the area is scarce), - extend the time span of figure 3b so that the entire production history could be visible (figure 3 of this answer).

*"In several places in the manuscript (including the abstract) the authors point on the subsidence bowl to be centered above the area where horizontal production wells are located. There is neither discussion nor reference to support why horizontal production wells should increase the subsidence. We can find in existing reports (not cited in the manuscript) a one order magnitude change in production rate on that period. Such a more quantitative information should be discussed in the context of subsidence rate change over time."*

Again, we would like to point out that very little bibliography is available on this particular oil field, and on the Myzeqeja plain in general. This is also the reason why we believe our original study could be of interest for a broad audience. If a reference is missing, we would be glad to read it and include it in the bibliography section. We have based our description of the oil field structure on the reports available on the Banker's company web site (Bankers report 2010, 2015) that helped us to indicate on Fig.3a of the main text three subzones of the field where different extraction techniques have been used since 2009. We agree with the reviewer that describing the area of maximal subsidence only as hosting horizontal wells may cause confusion and could suggest a relationship between horizontal wells and subsidence that is not supported by the observations. This area, plotted in green in Fig.3a and in bold black in others, also coincides with the highest density of wells observed from satellite images. Waterflooding recovery technique is also used there (Bankers reports). We think that the spatial relationship between the subsidence peak and this intensely operated area is an important observation that sheds some light on the causal link between both phenomena. Because we do not have access to individual well location and production history, we are not able to conclude on whether the subsidence could be associated with the extraction

technique by itself or with the volume of extracted fluid only.

As already said, we will provide a longer version of figure 3b yet showing the production evolution in BOPD for 1990 to present. As stated in the text, oil production is steadily and rapidly increasing since 2009.

*"Also, it appears there are problems with how the oil reservoirs are described in the paper (100m-2 km depth) when the available literature points on "Measured drilling depths are typically 2,500 meters, with 500- to 700-meter horizontal legs." (e.g., Mazerov, 2011). Also, Weatherill et al. (2005) report on "The two separate fields (Patos in the south and Marinza in the north) have productive sands at different depths, i.e., 0 - 1200 m for Patos and 1200 - 1800 m for MA". These differences are not discussed in the manuscript."*

Some discrepancies in the depth of the reservoirs indeed exists in the literature, mainly due to the fact that these formations are gently dipping north ($\sim 10°$) away from the eroded anticline trap located in the "core" zone of the field, close to the area of maximum subsidence. The papers cited in the main text (Silo et al., 2013; Prifti and Muska, 2013) are consistent with the informations given in the Bankers Corporate Presentation (2015) stating that well's depth varies from 300 to 2000m, or in Weatherhill (2015) "The productive sands under cold production occur across a gross interval of 250 m at a depth range of from 875 to over 2000 m." The cross section provided in Sejdini et al. Fig.9 also confirms this range of depth. It appears that in average, in the intensely operated area, wells are $\sim$1500m depth. However, we reckon that the caption of Figure 2d is misleading. We will clarify the vertical scale of this cross section and complete the description of the oil field in the text.

The terminology used to describe the oil field by itself depends on the author. Though, in general, as stated by the reviewer, one distinguishes two distinct parts in the Patos Marinza field : the Marinza part North of 40.7°N roughly, and the Patos one south of 40.7°N. The Banker's development plan since 2009 has mainly focused on the Marinza

zone. This is why we chose to focus on it on Figure 3a of the main text and explain why the oil field contour is different from previous figures where the entire oil field is shown (that should also answer to one reviewer comment just below).

*"Hydrocarbon field locations and numbers for other nearby oil-gas reservoirs do not overlap with other existing maps ( e.g., Mezini and Musai, 2012, Velaj SPE, 2015)."*

We will add numbers referring to names of the main fields represented in Fig.1. This map is consistent with the one published by Velaj et al. 1999 (fig.2 "Thrust Tectonics and the Role of Evaporites in the Ionian Zone of the Albanides").

**3    crude use of seismological catalogs unbalances the detailed technical analysis for the interferograms**

*"The well-constrained technical results for subsidence are discussed against poor seismicity and production history and without the proper fluid manipulation context, concerning the cited literature (most cited references relate to seismicity driven by wastewater disposal which is not the case in the submitted manuscript). The lack of production information and the misused of seismicity catalogs weaken the central message of the manuscript. Most of the discussions on the possible anthropogenic seismicity are flawed in several ways."*

As pointed out by the reviewer, the core of this study is the detection, quantification and modeling of the surface subsidence in the basin and above the oil field in particular. As stated in the manuscript, the seismic catalogs available in the area do not provide sufficient historical records nor register small enough magnitude to conclude on a possible induced seismicity in the last years. We reckon that some sentences of the main text are misleading and may appear too affirmative and that some complementary analysis can be done on the seismic catalogs to assess most of the reviewer's remarks. However, we would like to stress the fact that this paper has no ambition to

be a seismological study and rather brings new and independent InSAR observations of the ongoing ground deformation over the oil field.

*"First, there is no information on how the regional earthquake catalog is built on or selected by merging existing catalogs (USGS and EMSC)."*

In Fig.3 caption, the sentence : "Seismic events since 1950 are color-coded based on their occurrence time and sized depending on their magnitude (CSEM-EMSC; USGS, before and after 2004, respectively)." is wrong. It should be corrected to state that we use USGS catalog for the period pre-2004, and CSEM-EMSC catalog after 2004. We choose to use the more complete EMSC catalog for most recent seismicity since event location is often more precise than USGS for Europe, and its completion magnitude is lower (see above). However, no further data selection has been done.

*"Second, the lack of threshold values (i) for selecting an earthquake at a given distance from the reservoir to be a triggered event or a tectonic event and (ii) for a magnitude completeness value over time, prevent any robust analysis of space-time seismicity patterns."*

If we reckon that interpreting the increase of seismicity plotted in Fig.3b for the area requires a more thourough analysis of the completeness magnitude and of the regional seismic rate over a longer time-span (see following answers), we fear there may be a misunderstanding : we have no ambition to say whether an event is triggered or not, nor are we claiming to distinguish the triggered events in the main text. Nonetheless, the fact that seismic events in the last years have increased the population questioning about a potential impact of the oil extraction on its environment should not be ignored. Accordingly, this question would deserve a more specific investigation in the future.

Fig.3b (that is discussed by the reviewer) shows only a plot of cumulative moment over time on the restricted area ranging from 19.6 to 19.7°E and from 40.65 to 40.8°N. We previously chose this area as the most restrictive encompassing the oil field, and as suggested by the reviewer, we agree that we should compare this rate with the seismic

rate observed elsewhere and over a longer time-span. Nevertheless, our point is not to conclude on the induced or natural nature of the recorded seismicity.

We propose to clarify our reasoning in the revised version of the paper, based on the following arguments :

- We combine both USGS and EMSC catalogs to make a brief analysis of the regional and local seismicity in order to keep a magnitude completeness around 3.5 for the 1990-2019 time span (see figure 4 of this answer that will be added as supplementary figure in the revised version)

- We will discuss more carefully the history of both production and regional seismicity based on figure 5 of this answer. We calculate the cumulative moment released since 1939 (start of the Patos-Marinza oil field operation) by moderate earthquakes (Mw 3 to 6) on circular areas of 50 km and 15 km around the center of the oil field, taken as the maximum subsidence point (19.6492°E, 40.7395°N). The 50 km radius area encompasses the whole Myzeqeja plain and therefore remains n the same tectonic context. Before 1976, these curves have to be taken with caution since the completeness magnitude was around 4 to 5 (see figure 4 of this answer). On the long term on the large 50 km-wide area, no correlation appears between the production rate and the released seismic moment and no significant increase in the seismic moment rate is observed in the past years.

- In order to better understand the recent history of the oil field, we propose a new version of figure 3 of the submitted version of the manuscript presented in figure 6 of this answer. In figure 6a, the Mw 3.5+ 2013 earthquakes that stirred concern among the local population are contoured in bold black so that they could be easily identified. All of them are within 10 km from the oil field contour presented in this figure. Figure 6b has been enlarged in time (1990-2019) and the cumulative moment curve is now calculated over 15 km radial distance from the center of the oil field (see above). The 1997 Mw 4.8 event that did not appear on the previous version of this figure is now

associated to a clear jump in the released moment curve. On this time-span, the cumulative moment rate seems to increase after 2009, though, as pointed out by the reviewer and illustrated in figure 5 of this answer, this apparent change in moment release rate is not significant based on the longer term evolution.

- Finally, we will complete the bibliography with the references suggested by the reviewer that are focused on oil-field induced seismicity.

Fig.1 : LOS velocity map (ascending track) together with the main known faults and the location of major oil and gas fields (with names indicated in small capital letters). The fields contours have been extracted from the IHS map of Albanian ressources of february 2001. The contour of the Patos-Marinza oil field is larger than the one plotted in figures 1 to 6 of the main text, that has been extracted from a more recent reference [Bankers corporate presentation 2015].

Fig.2 : Same as figure 1 for descending track.

Fig.3 : History of production of oil (plain line in BOPD) and gas (ticked line, in BOED) in the Patos-Marinza field (BKC, 2015).

Fig.4 : Test of the Gutemberg-Richter relationship over the southern Balkans area (top) and Albania (bottom) for different time-spans and catalogs. Inset map in the top right corner of the graphs shows the region over which the seismicity is considered. Since 1990, the completeness magnitude can be safely estimated between 3 and 3.5. Interestingly, EMSC catalog is much more complete than USGS catalog for Albania on the 2004-2018 time span.

Fig.5 : a- History of production of oil (plain line in BOPD) and gas (ticked line, in BOED) in the Patos-Marinza field (BKC, 2015). b- Cumulative seismic moment released since 1939 based on the combined USGS-EMSC catalog over a 50 km radius area around the oil field (plain line) and a 15 km radius area around the oil field (dashed line).

Fig.6 : a- Same caption as figure 3 of the submitted version except that Mw 3.5+ 2013

earthquakes referred in the text are contoured in bold black together with the Mw 4+ 2016 earthquake. b- Zoom on the 1990-2019 cumulative seismic moment released in a 15 km radius area around the oil field center (black) together with the oil-field production evolution (purple).
* * *
Fig. 1.

[Figure]

The map shows LOS (mm/yr) deformation with legend including oil field, gas field, Normal faults, Strike-slip faults, Thrust faults, Main cities, and Main rivers. Labels on the map include DIVJAKA, Lushnje, PEKISHTI, KUCOVA, Fieri, PATOS-MARINZE, Marinza, Zharrez, POVELCA, FRAKULJA, VISOKA, CAKRAN-MOLLAJ, BALLSH-HEKAL, DRASHOVICA, Vlora.

Coordinate labels: 19.2, 19.4, 19.6, 19.8, 20.0 (longitude); 41.0, 40.8, 40.6 (latitude).

**Fig. 2.**

Primary recovery phase by APOC and national company

Transition period
Political instability

Oil field
reactivation
Banker's

oil production (BOPD) or gas production (BOED)

time

25000

20000

15000

10000

5000

0

1939   1949   1959   1969   1979   1989   1999   2009

**Fig. 3.**

[Figure]

**Fig. 4.**

**Fig. 5.**

[Figure]

**Fig. 6.**

---

## Referee Comment (RC2) · Simon McClusky (Referee) · 13 Nov 2019

While the analysis and modelling of the InSAR LOS deformation the Patos-Marinza oil field provides good quantitative evidence that extraction of hydrocarbons is leading to deformation, the analysis of seismic moment release and the limited production data are to simplistic to support the vaguely worded conclusions that extraction has lead to increased seismicity in the region.

To support the hypothesis that extraction has induced seismicity, the MS needs to much

more carefully statistically evaluate the seismicity data relative to the deformation results. It is not clear how the radii for computing moment release was chosen or how this chosen region is related to the deformation model derived. Ie does the model produce significant stress change across the region from which the seismic moment is summed. Even some simple analysis of what the background tectonic rate of stress/strain accumulation is, and how this might be manifest in seismic moment release. Is this sort of seismicity clustering occurring elsewhere in Albania?

As it currently stands in my opinion this paper is far to speculative in section 6 "Conclusions & implication for local seismic hazard" there are statements made about stress changes that have no basis, because the stress change calculations were not presented in the paper? There is simply not enough direct evidence presented in the paper to support the conclusions drawn, all be they vague..
* * *

---

## Author Comment (AC3) · 11 Dec 2019

We thank reviewer 2/ (Simon McClusky) for his comment on our paper "Subsidence associated with oil extraction, measured from time-series analysis of Sentinel-1 data : case study of the Patos-Marinza oil field, Albania".

The main concern expressed in this review, i.e. the fact that more detailed analysis on the seismological data are required to conclude on whether induced seismicity is occurring in the oil field, is in accordance with the one pointed out by reviewer 1. As

already stated in our reply to reviewer 1, we agree that some sentences of the submitted manuscript on induced seismicity may appear too affirmative and should be toned down. We also propose a more careful analysis of the local seismicity. Nonetheless, we reiterate that local seismicity remains poorly known due to sparse local and regional seismic networks (see figures 4, 5 and 6 of our reply that we wish to include in the revised version of the manuscript). Our main purpose on this paper is to bring new observations that clearly show relationship between surface deformation and oil extraction, the revised version of the manuscript will insist more on this point. The link between oil extraction and seismicity is only discussed in the paper as one hypothesis that remains to be tested but is out of the scope of this study.

That being said, we note that Reviewer 2 says "the analysis and modelling of the InSAR LOS deformation the Patos-Marinza oil field provides good quantitative evidence that extraction of hydrocarbons is leading to deformation", i.e. that the core of our paper is convincing and not subject to caution. We would like to stress the fact that our aim is to present original evidences of the subsidence of the whole Myzeqeja plain, and of a local high-rate subsidence that is spatially correlated to the center of the Patos-Marinza oil field. In the submitted version of the manuscript, we are not claiming that the observed seismicity in the area until 2004 has been induced but that such localized and strong subsidence rate as the one seen by InSAR may be associated to significant pressure changes at depth. Since this point has been questioned by both reviewers, we reckon we should rephrase parts of the paper and add new supplementary figures.

"It is not clear how the radii for computing moment release was chosen or how this chosen region is related to the deformation model derived. Even some simple analysis of what the background tectonic rate of stress/strain accumulation is, and how this might be manifest in seismic moment release. Is this sort of seismicity clustering occurring elsewhere in Albania ?"

To answer this question, we kindly refer to figures 5 and 6 of our previous answer to reviewer 1. We propose to add a figure showing the evolution of the moment release

since 1939 when the production started. We also modify current figure 6Âă: it is now broadened in time (1990-2019) and the cumulative moment curve is now calculated over 15 km radial distance from the center of the oil field. Over the course of the past century, no obvious correlation can be found between seismically released moment and the oil production when looking both at seismicity over a 50 km radius representative of the whole Myzeqeja alluvial plain, and over a 15 km radius more representative of the oil field potential influence (Figure 5). However, looking at the seismic moment release since 1990 in this restricted 15 km radius area shows an apparent increase in the moment rate starting in 2009 (Figure 6). Although we recognize that the statistical significance of this slight change is uncertain given the moderate seismicity background and the poor catalogue, and in any case impossible (given our current state of knowledge) to relate to the resurgence of oil extraction activity around 2005, we suggest that this relative increase in seismicity may explain the local population concern.

"As it currently stands in my opinion this paper is far to speculative in section 6 "Conclusions & implication for local seismic hazard" there are statements made about stress changes that have no basis, because the stress change calculations were not presented in the paper? There is simply not enough direct evidence presented in the paper to support the conclusions drawn, all be they vague."

We did not conduct stress calculations because we have very few information on the reservoir structure, initial pressure, wells locations etc. The first order modeling we do propose in section 5 shows that we can explain the big picture of the observed LOS motion with standard simplified compaction elastic or poro-elastic models, but we are well aware of the fact that much more refined models should be conducted to model pressure changes in the reservoir. This is what we state at the end of section 5 "Finally, to properly assess the net volumetric change of the reservoirs, one would need to build a 3D finite elements model of the media taking into account its physical properties, geometry, pressure history, etc., i.e. data we do not have access to at the moment."

Finally, Reviewer 2's feeling that our conclusions are not supported by enough evidences may stem from the fact that most of the focus was deflected in the online discussion of our paper toward the debate of whether seismicity was induced by oil extraction or not, whereas surprisingly little attention was given to the main contribution of our work. We recall that our paper is focused on new observations of intense and undocumented subsidence over the largest Albanian oil and gas field. Our work opens new issues about the nature of the seismicity observed in the area that we think should not be eluded, even if we do agree caution is required.

We start section 6 saying that "Unfortunately, local seismological arrays are still too sparse to provide accurate locations and focal mechanisms of the small magnitude earthquakes occurring in the vicinity of the Patos-Marinza oil field, a necessary prerequisite to conclude on the induced nature of the recent increase in local seismicity", that seems to be in agreement with the reviewer's point of view.

We conclude section 6 saying that "Whether the stress changes associated with the oil extraction in the Patos-Marinza field may be sufficient to trigger earthquakes on underlying faults is therefore an open question that challenges the seismic-hazard assessment in the area. Denser and more precise seismic catalogues, together with longer InSAR or other geodetic time-series to monitor the spatio-temporal evolution of the deformation, as well as detailed knowledge of the wells injection history and reservoir properties, will be essential to explore this issue further on." that is also a claim for caution. In order to answer both reviewer's concern, we will rephrase section 6 in order to insist more on our robust and original observations of land motion and tone down some discussion about potentially induced stress changes.

As a conclusive remark, we would like to insist on the fact that the seismic catalogue is simply not complete enough to support or reject the induced seismicity hypothesis. This situation will remain the same for a long time, if not forever. Our paper provides compelling observations of the impact of oil extraction on the vertical displacement of the surface, suggesting that underground effects (in terms of pressure changes) are substantial. Instead of restraining our analysis to the geodetic data processing

and modeling, we enriched the paper by putting together a thorough review of recent changes in seismicity and oil extraction rates in Albania. We believe that this effort should be recognized as an attempt to elevate the level of the debate and point toward a research direction of obvious interest for seismic hazard assessment in the area. For this reason, we made our best effort to provide as much information as possible on our current state of understanding of the situation, including the main observations and the main questions pending on the subject. We claim that this effort will contribute to increasing the interest of readers on our paper and on the Solid Earth journal. Following both reviewer's suggestions, we think that the careful rephrasing of our discussion and conclusion sections will ensure that nothing is scientifically wrong in our assertions, while being sufficiently supported by data and prior knowledge of processes of induced seismicity to avoid excessive vagueness.

---

## Author Response (AR2)

Comments to the Author:

The MS is much improved. Some minor technical problems only before it is ready for publication.

1) You need to proof read the document carefully. There are still numerous grammatical and style errors that need to be addressed. as the topical editor I don't have the capacity to do copy editing for you.

The final version of the manuscript has been proofread by a native english speaker (Tara Shreve) that is acknowledged at the end of the manuscript.

2) There is no length scale on any of the figures with maps in them. This needs to be rectified.

Length scales have been added to figures 2 and 5 where they were lacking. The other maps go together with profiles that are labeled in meters and therefore provide a scale to the reader.

3) You need to mark on fig 2 and/or figure 4 and figure 6 the 15 and 50 km radius zones in which you are analysing the seismicity and mention what the zones are in the figure captions. Currently it is impossible for the reader to see how these zones relate to the deformation field or plotted seismicity. Also it is not clear how these radii relate to the physical hydrocarbon extraction zones. Figure 4 might need to be enlarged in spatial coverage to fit the 50 km radius zone? Maybe this will have to be shown on fig 2?

We choose to add the 15 km and 50 km contours on panel C of figure 5 that presents the large scale velocity maps and we refer to these contours in Figure 3 caption.